# SCRREAM : SCan, Register, REnder And Map: A Framework for Annotating Accurate and Dense 3D Indoor Scenes with a Benchmark

**HyunJun Jung**
Technical University of Munich
`hyunjun.jung@tum.de`

**Weihang Li**
Technical University of Munich

**Shun-Cheng Wu**
Technical University of Munich

**William Bittner**
Technical University of Munich

**Nikolas Brasch**
Technical University of Munich

**Jifei Song**
Huawei Noah's Ark Lab

**Eduardo Pérez-Pellitero**
Huawei Noah's Ark Lab

**Zhensong Zhang**
Huawei Noah's Ark Lab

**Arthur Moreau**
Huawei Noah's Ark Lab

**Nassir Navab**
Technical University of Munich

**Benjamin Busam**
Technical University of Munich
`b.busam@tum.de`

## Abstract

Traditionally, 3d indoor datasets have generally prioritized scale over ground-truth accuracy in order to obtain improved generalization. However, using these datasets to evaluate dense geometry tasks, such as depth rendering, can be problematic as the meshes of the dataset are often incomplete and may produce wrong ground truth to evaluate the details. In this paper, we propose SCRREAM, a dataset annotation framework that allows annotation of fully dense meshes of objects in the scene and registers camera poses on the real image sequence, which can produce accurate ground truth for both sparse 3D as well as dense 3D tasks. We show the details of the dataset annotation pipeline and showcase four possible variants of datasets that can be obtained from our framework with example scenes, such as indoor reconstruction and SLAM, scene editing & object removal, human reconstruction and 6d pose estimation. Recent pipelines for indoor reconstruction and SLAM serve as new benchmarks. In contrast to previous indoor dataset, our design allows to evaluate dense geometry tasks on eleven sample scenes against accurately rendered ground truth depth maps. (`https://sites.google.com/view/scrream/about`)

## 1 introduction

Indoor tasks are heavily related to human activities. As a consequence, perceiving 3D indoor scenes is one of the major tasks in 3D computer vision. A series of indoor datasets [1–8] with different annotation methods has been introduced to the vision community to provide the data necessary for training and for performance evaluation. Such datasets, however, are generally capturing a scene with a depth sensor from a limited number of viewpoints, trying to avoid making any changes to the scene

38th Conference on Neural Information Processing Systems (NeurIPS 2024) Track on Datasets and Benchmarks.

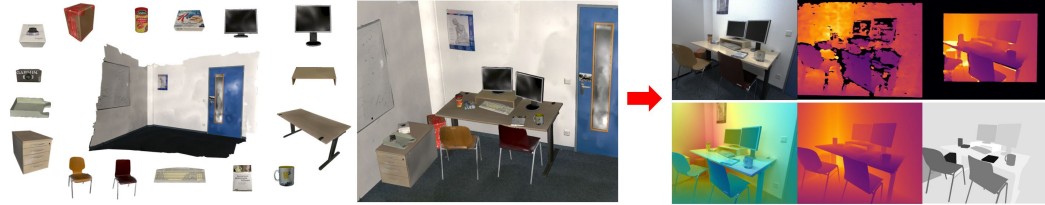

**Individually scanned high resolution meshes registered to the real scene**  **Real RGBD Sequence + Rendered Ground Truth**

Figure 1: Due to the typical acquisition pipelines, traditional indoor 3D datasets can provide incomplete meshes for their scenes with missing structures and holes. Our dataset annotation pipeline, in contrast, starts from scanning individual objects in an high resolution manner and then registers them to the real scene and real camera sequence allowing highly detailed ground truth rendering for dense 3D vision tasks.

to simplify registration and fusion. While this way of scanning can speed up the dataset capturing process and allows for fast large scale data creation, it has the drawback of potentially acquiring incomplete meshes due to e.g. line of sight problems or lack of sensor accuracy.

On the other hand, [9] showed that scanning all objects individually and consecutively registering them into the scene can ensure dense and complete annotations for all objects. In turn, they can be used to render dense ground truth. However, using a robotic arm limits the scene setup as well as the camera trajectory to scenarios such as table-top scenes with cameras that cannot move freely due to the robot's limited joint operation range. Therefore we design a novel pipeline that replaces the tool tip based object registration with a highly accurate partial re-scan and registration procedure. We adapted feature matching based pose estimation leveraging synthetic views rendered from the registered model similar to [7] to replace external hardware based camera tracking (see Fig. 1).

With SCRREAM we follow the spirit of our previous 3D dataset projects PhoCal [10], Hammer [9] and HouseCat6D [11] that democratized advances in this field for smaller scale 3D perception and manipulation tasks by describing all pipeline parts and the hardware components in detail, enabling the community to apply the acquisition principle on their own hardware setups. We further provide the source code of visualization tools with detailed documentation to allow users to utilize the SCRREAM data for downstream tasks. The entire dataset is made available for the community[1] and we provide a benchmark for 3D geometric tasks using existing state of the art (SoTA) methods for indoor reconstruction & SLAM on our example scenes.

To this end, our contributions are :

1. We propose a dataset annotation framework that is capable of annotating fully dense indoor scenes. It can be applied to different indoor perception tasks.

2. We detail the steps of our annotation framework for future use and release an acquired dataset into the public domain with detailed documentation.

3. We create a new benchmark for scene level geometry tasks comprising indoor scene reconstruction, novel view synthesis and SLAM using accurately rendered dense depth ground truth as well as precise meshes.

## 2 Related Works

Early datasets obtain their annotations by using directly the depth sensor readings [1, 2], where invalid or empty pixels are manually annotated to provide dense 3D data for evaluation [1]. Later, the paradigm for annotation changed, i.e. the indoor scene is reconstructed by fusing a sequence of depth images obtained by multi modal camera trajectories [3–5] to produce more complete scene geometry. Depth and other 3d information is produced on the image plane by rendering the mesh using the camera pose. To improve the quality and density of the datasets, annotation methods that pre-scan the scene, record [6] and localize the camera are developed [7, 8]. Although this annotation technique

---

[1]Dataset link and toolkit can be found in `https://github.com/Junggy/SCRREAM`

Table 1: **Dataset Comparison.** Each scene in our dataset is annotated with individually scanned water-tight high resolution meshes with texture maps which are provided as digital twin assets. Thus, fully dense ground truth maps such as highly detailed depth and instance/class segmentation maps can be rendered. This serves as accurate benchmark data. We additionally provide scenes with reduced objects in the scene to provide object removal and scene editing setup that comprises of extra 9k frames.

| Dataset | Real Data | RGB Sequence | Polarization | Camera Pose | Scene Reconstruction Mesh | Digital Twin Asset | Class Segmentation | Instance Segmentation | Scene Editability | Free-Hand Camera | Accurate Depth GT | Photogrammetry Depth | Structured Light Depth | ToF Depth | Active Stereo Depth | Number of Scenes | Number of Frames |
|---|---|---|---|---|---|---|---|---|---|---|---|---|---|---|---|---|---|
| NYU [1] | ✓ | ✓ | | | | | ✓ | | | ✓ | | | ✓ | | | 464 | 1449 |
| TUM RGBD [2] | ✓ | ✓ | | ✓ | | | | | | ✓ | | | ✓ | | | 6 | > 10k |
| ScanNet [3] | ✓ | ✓ | | ✓ | ✓ | | ✓ | | | ✓ | | | ✓ | | | 1513 | 2.5M |
| Matterport3D [4] | ✓ | ✓ | | ✓ | ✓ | | ✓ | | | | | ✓ | | | | 2056 | 194k |
| Stanford2D-3D [5] | ✓ | ✓ | | ✓ | ✓ | | ✓ | | | | | ✓ | | | | 25 | 70k |
| Replica [6] | ✓ | | | | ✓ | | ✓ | | | | | - | - | - | - | 18 | N/A |
| ReplicaCAD [12] | | | | | ✓ | ✓ | ✓ | ✓ | ✓ | | | - | - | - | - | 111 | N/A |
| RoboTHOR [13] | | | | | ✓ | ✓ | ✓ | ✓ | ✓ | | | - | - | - | - | 89 | N/A |
| HAMMER [9] | ✓ | ✓ | ✓ | ✓ | ✓ | ✓ | ✓ | ✓ | | | ✓ | | ✓ | | ✓ | 13 | > 10k |
| ARKitScenes [8] | ✓ | ✓ | | ✓ | ✓ | | ✓ | | | ✓ | | | ✓ | | | 1661 | 450k |
| ScanNet++ [7] | ✓ | ✓ | | ✓ | ✓ | | ✓ | | | ✓ | | | ✓ | | | 5047 | 3.7M |
| Ours | ✓ | ✓ | ✓ | ✓ | ✓ | ✓ | ✓ | ✓ | ✓ | ✓ | ✓ | | ✓ | | ✓ | 11 | 7k(+9k) |

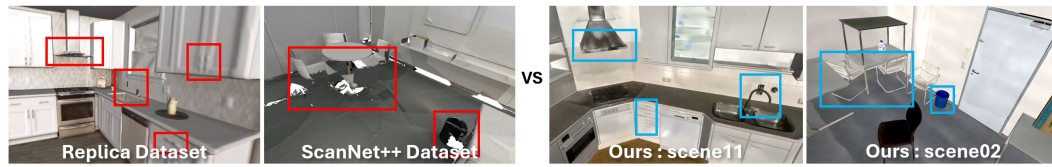

Figure 2: **Dataset Geometry Comparison.** Our dataset features high-quality, complete meshes of the scene. In comparison, commonly used datasets such as Replica [6] and ScanNet++ [7] suffer from over-smoothed or incomplete meshes (Zoom in for details). We include more in-depth comparisons in the supplementary material pdf and video file.

produces a significantly better mesh quality of the scene, it suffers from line of sight problems of the scanner as the scanning is done on a scene level. Ensuring every single object in the scene is complete is a non-trivial task. As a result, dense ground truth generated in this way suffers from holes and missing parts (Fig. 2). Therefore, scene-level reconstruction methods like neural radiance fields (NeRF) [14] variants or 3D Gaussian Splatting [15] evaluate their geometry only via qualitative evaluation on their depth prediction.

To ensure complete scenes, [12, 13] proposed synthetic data with high-quality CAD models created by artists. ReplicaCAD [12] proposed a synthetic version of Replica dataset [6] that contains room of Replica dataest but with different layouts with CAD models. Similarly, the Robothor dataset creates a modular room that contains different layouts from CAD models based on IKEA furniture. These approaches ensure the scene's completeness and the dataset's scale. However, as the aim of the dataset is instead a robotic simulation, the data is not focused on realism and thus suffers from sym and real domain gaps. HAMMER [9] constitutes an annotation method that uses a robotic arm to annotate pre-scanned meshes as well as a camera trajectory. In contrast to other datasets, all meshes are pre-scanned individually to make sure there is no missing part such that holes in the area of interest cannot occur. This results in annotated scene from which highly accurate depth rendering can be acquired without any artifacts. These can be used as absolute ground truth for depth prediction tasks. However, the robotic arm's range of motion is limited such that the acquisition process is capable of annotating only table top scenarios with small camera motion. HouseCat6D [11] replaces

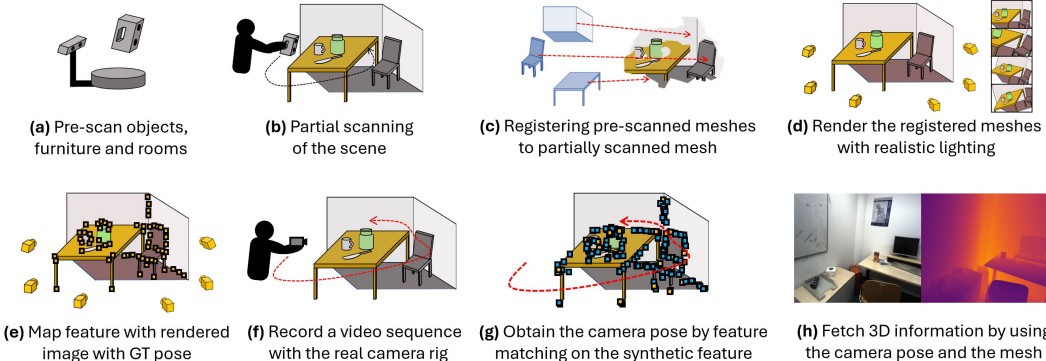

**(a)** Pre-scan objects, furniture and rooms

**(b)** Partial scanning of the scene

**(c)** Registering pre-scanned meshes to partially scanned mesh

**(d)** Render the registered meshes with realistic lighting

**(e)** Map feature with rendered image with GT pose

**(f)** Record a video sequence with the real camera rig

**(g)** Obtain the camera pose by feature matching on the synthetic feature

**(h)** Fetch 3D information by using the camera pose and the mesh

Figure 3: **Framework Pipeline Overview.** Our pipeline follows the SCRREAM scheme for annotation. **(a) SC**an : Scanning the individual objects in the scene, **(b,c) R**egister : Place objects in the scene, scan the scene partially and register the pre-scanned meshes, **(d) RE**nder : Render the synthetic images, **A**nd **(e-g) M**apping : Map 3D features of synthetic image with camera poses, record the real image sequence and obtain the camera pose via feature matching. Once the camera poses are obtained, we can extract or render the 3D information via transforming the meshes into the camera frame as shown in **(h)**.

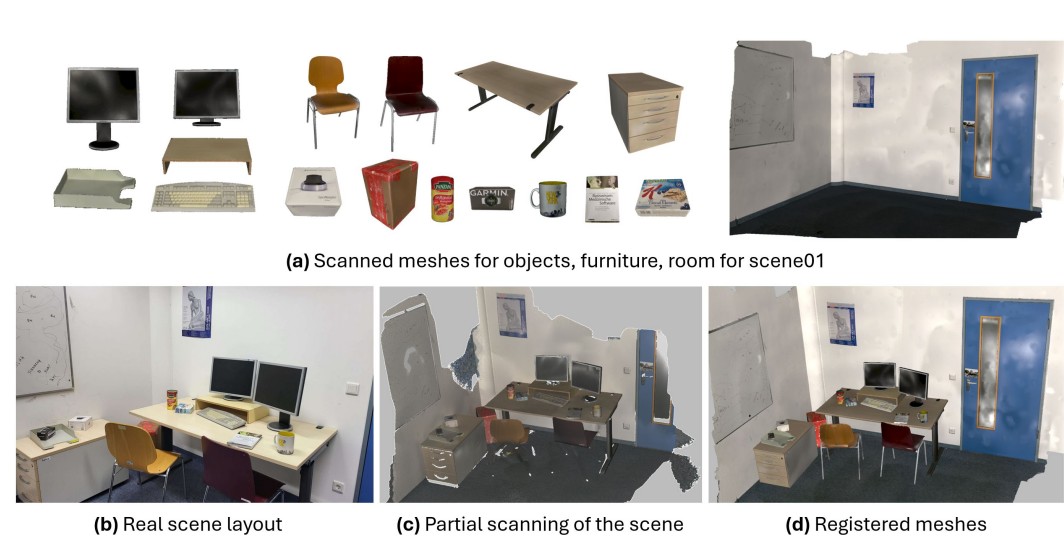

**(a)** Scanned meshes for objects, furniture, room for scene01

**(b)** Real scene layout

**(c)** Partial scanning of the scene

**(d)** Registered meshes

Figure 4: **Example for the Scan and Register Step.** **(a)** We pre-scan all meshes in the scene before setting up the scene. This ensures that all objects and furniture are scanned in a high quality, water-tight manner. **(b)** Then we place the furniture in the room to setup the scene and **(c)** scan the entire scene (Note that the scene is not scanned completely), such that we can **(d)** register all pre-scanned meshes to the scene layout via manual correspondence selection followed by ICP.

the robotic arm with an external tracking device for better flexibility and wider camera motion. While it improves both scene and camera pose diversity significantly, the dataset cannot achieve true free hand-held camera trajectories due to line of sight problem of the tracking device. Moreover, the tracking camera set up in the background of the scene makes the dataset less suitable for indoor reconstruction purposes. In this work, we combine the two methods for annotation [9, 11] and [7, 8], and achieve a fully dense annotation pipeline of scenes as well as free hand-held camera trajectories.

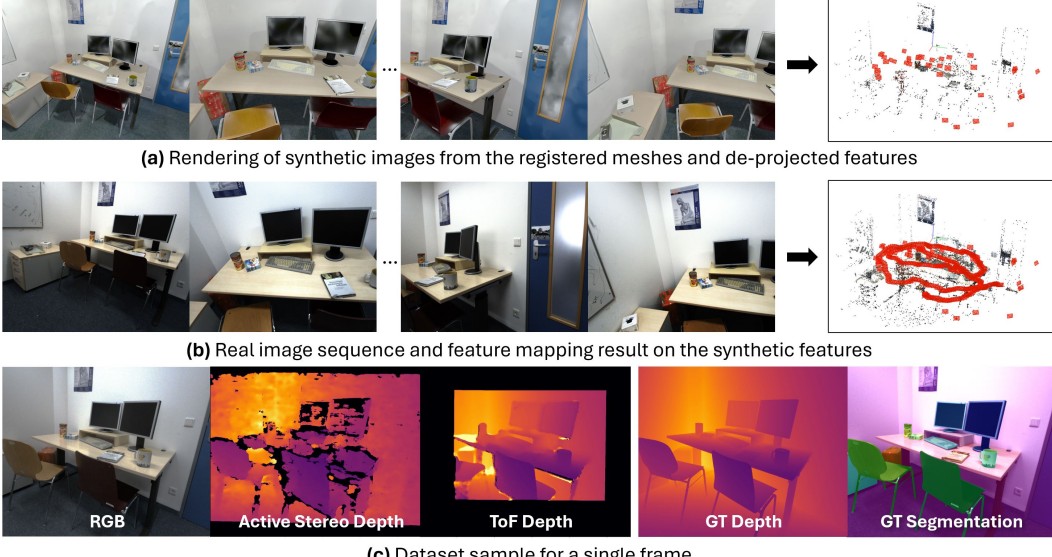

(a) Rendering of synthetic images from the registered meshes and de-projected features

(b) Real image sequence and feature mapping result on the synthetic features

RGB    Active Stereo Depth    ToF Depth    GT Depth    GT Segmentation

(c) Dataset sample for a single frame

Figure 5: **Example for the Mapping Step and Qualitative Evaluation.** Mapping starts with **(a)** generating realistic synthetic renderings from the registered mesh. Once the images are rendered, features are matched and de-projected to 3D. Then **(b)** the real sequence is acquired, and features are extracted to match with synthetic features to obtain the camera poses. These camera poses allow us to transform the camera frame to the mesh frame such that dense 3D annotations **(c)** can be rendered into the image as ground truth. We show rendered instance masks as well as depth on the real image frame to illustrate the quality of our ground truth.

## 3 Dataset Acquisition Method

Our goal is to setup a pipeline that is capable of annotating the fully dense scene as well as image sequence recorded by a free hand-held camera with reliable camera poses. Our pipeline follows the **SCRREAM** scheme : **SC**anning the complete meshes, **R**egistering the meshes to the scene, **RE**nder the synthetic scene **A**nd **M**apping the video sequence to a synthetic scene to obtain the camera pose. In this section, we explain in more detail each part of the SCRREAM pipeline.

**Scan.** In the scanning step, we scan the entire empty room, furniture and small objects in the scene separately to obtain the complete meshes such that all the objects and furniture in the scenes are complete regardless of line of sight of the scanner or camera trajectory. We use an EinScan-SP (SHINING 3D) for scanning small household objects and an Artec Leo (Artec 3D) for scanning the furniture and room. For the small household objects and furniture, we make sure the scanned meshes are water tight and have high quality texture maps. The scanning step corresponds to Fig. 3 (a) in the pipeline, and examples for scanned meshes are shown in Fig. 4 (a).

**Register.** Once the scanning is done for all objects, the furniture and room, we create the scene by placing the furniture and objects in the room. The scene is then scanned as a whole with the hand-held scanner to provide the sparse mesh that can be used to register the pre-scanned meshes to the scene similar to [9]. Initial registration is done by selecting 3-5 pairs of corresponding points between the pre-scanned mesh and the sparse mesh. This is further refined using ICP [16]. The register step corresponds to Fig. 3 (b),(c). An example of a partial scan and the corresponding registered meshes can be found in Fig. 4 (c),(d). For convenience, we use a commercial software (Artec Studio 17 Professional) for this registeration step.

**Render.** The registered scene with the pre-scanned meshes are then used to render the scene in a realistic manner. We use Blender4.0 [17] for rendering the scene by importing all registered meshes and placing the light source close to the real scene. We render around 50-100 frames that can cover many views of the scene that can be used for the next mapping step and save the corresponding

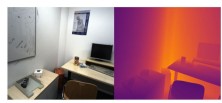
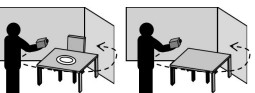
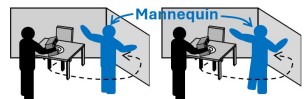
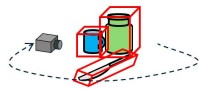

**(a)** Indoor Reconstruction and SLAM Dataset

**(b)** Scene Editing Dataset

**(c)** (Semi) Dynamic Human Reconstruction Dataset

**(d)** 6D Pose and Object Detection Dataset

Figure 6: **Dataset Variation Overview.** Depending on the scene setup, our annotation framework can produce different variants of ground truth. (a) rendering depth for indoor reconstruction and SLAM; (b) capturing scene with less objects produces ground truth for scene editing; (c) capturing scene with a mannequin in stop-motion for semi dynamic human reconstruction; (d) capturing a 360-degreee video around the objects for a 6D pose and detection dataset.

camera poses and intrinsics as a pair. The rendering step corresponds to Fig. 3 (d). Examples of rendered images are shown in Fig. 5 (a).

**Mapping.** The mapping stage maps the real image sequence to the rendered sequence that is obtained by the previous step. To obtain the real image sequence of the scene, we use a multi-modal camera rig that is composed of an RGB+P sensor (Lucid Phoenix with Sony Polarsens), an active stereo depth sensor (intel RealSense D435) and an I-Tof depth sensor (Lucid Helios). Extrinsic calibration is obtained from each depth sensor to the RGB sensor using the calibration scheme similar to [11]. We first place a ChArUco board and capture image sequences from all cameras to obtain sequences with trajectories $T_{camX \rightarrow board}$. For depth sensors, infra-red images are used to capture the ChArUco board. Then we align the trajectories from Depth sensors to RGB sensor using Horn's method [18]. The alignment matrix is an extrinsic matrix between RGB and the depth sensor. Once the extrinsic matrices are obtained from each depth sensor to the RGB sensor, the corresponding depth image is aligned to the RGB image using its own sensor depth and their extrinsic. None that we use noisy sensor depth to align depth images to the RGB image, which leads extra noises in the final depth image. We design this way specifically as this is how the real depth sensor produces an aligned depth image to its RGB image. All three cameras are synchronized by a hardware sync signal generated via a Raspberry Pi. We include more hardware details in the supplementary material.

Once the real image sequence is obtained, our task is to obtain the camera pose relative to the center of the registered scene meshes. We modify the off-the-shelf Structure from Motion (SfM) method COLMAP [19, 20] into two stages (two-stage mapping) to obtain the real camera poses from the synthetic images paired with camera pose and intrinsics. We first match the features from synthetic images and de-project them into 3D by using the paired camera poses and intrinsics. We then extract features of real images and match them to the synthetic image features and the corresponding camera poses are optimized to map the de-projected real image features onto de-projected synthetic features. The mapping step corresponds to Fig. 3 (e)-(g). An example of a real image sequence and mapped features with the retrieved camera pose is shown in Fig. 5 (b).

When the camera pose is obtained, we can map the available 3D information onto the camera frame by using the camera pose and meshes to annotate the scene frames with, e.g. dense depth (see Fig. 3 (h)), object poses, segmentation masks, surface normals, bounding boxes etc. We show possible variations of the dataset acquisition with the SCRREAM annotation framework in Sec. 4.

# 4 Dataset Variations

Our SCRREAM annotation framework is specialized in annotating fully dense meshes of a given scene with a freely moving camera. Depending on how the scene is set up, our pipeline can be used to annotate ground truth on data for different types of tasks (Fig. 6), such as indoor reconstruction and SLAM, object removal and scene editing, human reconstruction and 6D pose estimation. For each of the four variants, we provide publicly available data and provide detailed documentation in the supplementary material.

**Indoor Reconstruction and SLAM Dataset.** In this part, we consider indoor reconstruction and SLAM as the major task of our dataset. The dataset can be obtained by following the annotation pipeline described above. Dense depth maps and instance masks are rendered with camera poses and annotated meshes of the scene as shown in Fig. 5 (c). We acquire **eleven scenes** for this task (see

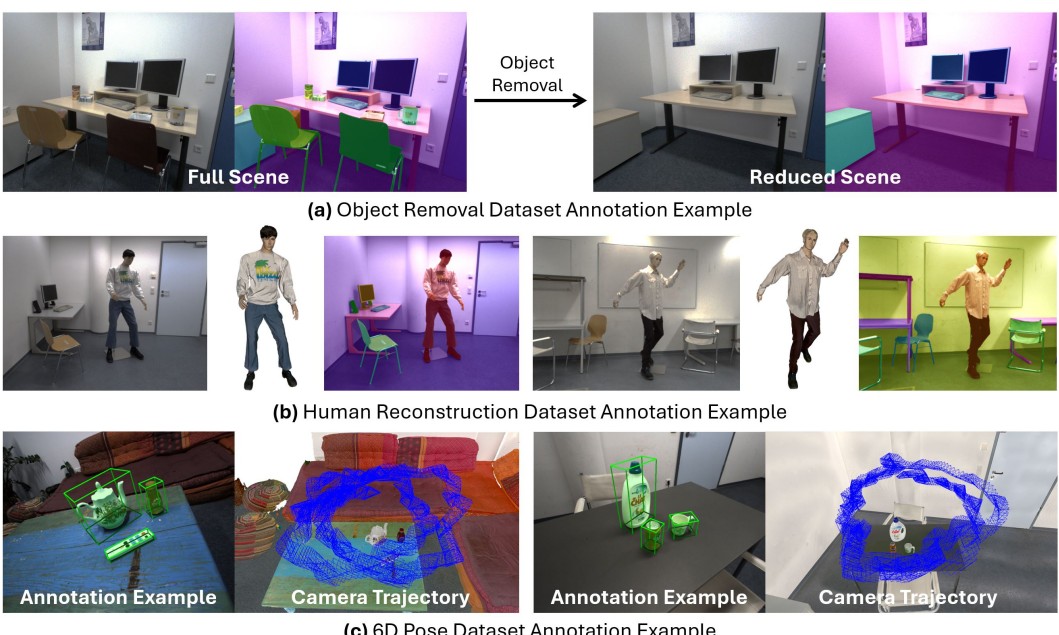

**(a)** Object Removal Dataset Annotation Example

**(b)** Human Reconstruction Dataset Annotation Example

**(c)** 6D Pose Dataset Annotation Example

Figure 7: **Dataset Variation Example. (a)** Our scene editing and object removal dataset provides mask and geometry annotation with and without the selected objects. **(b)** Our human reconstruction dataset is semi-dynamic using a stop motion mannequin, but each frame is fully scanned to provide the high quality reconstruction ground truth. **(c)** The 6D pose dataset from our pipeline can produce high quality annotation and 360 degree view of objects without any markers.

Fig. 10 for more details) and provide the benchmarks that evaluate the quality of view synthesis and 3D geometry tasks using recent approaches in Sec. 5.

**Object Removal & Scene Editing Dataset.** Object removal and scene editing datasets can be created by capturing the scene multiple times with less objects appearing in the scene (reduced scene). The advantage of using our pipeline is that one can easily exclude the object meshes that need to be removed from the scene and render the correct depth to evaluate the geometry. Also running the mapping on both camera trajectories with and without the target objects ensures both camera poses to be using the same reference. We recorded extra trajectories in this reduced setup for **eight selected scenes** from the eleven Indoor Reconstruction and SLAM Dataset scenes. Examples are shown in Fig. 7 (a).

**Human Reconstruction Dataset.** Human reconstruction datasets can be divided into two categories based on their capture setup. While the first set of datasets captures the human in a motion capture setup [21–23] (i.e. a static multi-camera recording studio with a human performer in the middle), the second acquisition setup focuses on real scenes with changing backgrounds and uses a simple hand-held camera [24] that can also be used outdoors. The former datasets have the advantage of being capable of capturing high quality annotation, but they lack realism in the scene as the background is the camera acquisition sphere. The latter lack the accuracy while they have the advantage of a realistic background. In our case, we use a mannequin with multiple joints to create a stop motion video, but scan and annotate the mannequin each frame with our annotation pipeline. This allows to create a human dataset that contains realistic backgrounds as well as high quality mesh annotations for both background and human. **Two sample scenes** are collected to showcase the high quality human annotation on each frame in the sequence as well as the realism of the background. We also annotated SMPL [25] parameters using the RVH Mesh registration repository [26, 27]. Example scenes are shown in Fig. 7 (b).

**6D Pose Estimation Dataset.** By nature of our annotation, the pipeline co-stores object poses and camera trajectories from the scene origin. Concatenating the two poses in the right order produces 6D object pose in camera coordinates. Using our pipeline allows for better camera coverage (e.g.

Table 2: **Quantitative Evaluation for Novel View Synthesis.** In addition to the photometric evaluation (RGB), our indoor benchmark provides ground truth depth. This allows the evaluation on the rendered depth to study each method's performance on the geometric reconstruction (depth). We differentiate between NeRF methods (upper part) and Gaussian Splatting methods (lower part).

| Evaluation | RGB | | | Depth | | | | | |
|---|---|---|---|---|---|---|---|---|---|
| Method | PSNR↑ | SSIM↑ | LPIPS↓ | RSME↓ | Abs Rel↓ | Sqr Rel↓ | $<1.25^1$ ↑ | $<1.25^2$ ↑ | $<1.25^3$ ↑ |
| NeRFacto [30] | 22.645 | 0.765 | 0.343 | 1.244 | 0.645 | 1.231 | 0.429 | 0.597 | 0.726 |
| Depth-Facto [30] (AS) | 24.502 | 0.786 | 0.324 | **0.218** | **0.079** | **0.059** | **0.968** | **0.981** | **0.988** |
| Depth-Facto [30] (ToF) | 24.540 | 0.788 | 0.323 | 0.336 | 0.093 | 0.149 | 0.926 | 0.943 | 0.957 |
| Zip-NeRF [31] | **28.315** | 0.783 | **0.259** | 0.493 | 0.245 | 0.189 | 0.546 | 0.825 | 0.887 |
| Gaussian-Splatting [15] | 25.943 | 0.801 | 0.328 | 0.526 | 0.218 | 0.174 | 0.589 | 0.806 | 0.883 |
| Mip-Splatting [32] | 25.925 | **0.802** | 0.327 | 0.532 | 0.223 | 0.178 | 0.594 | 0.794 | 0.875 |

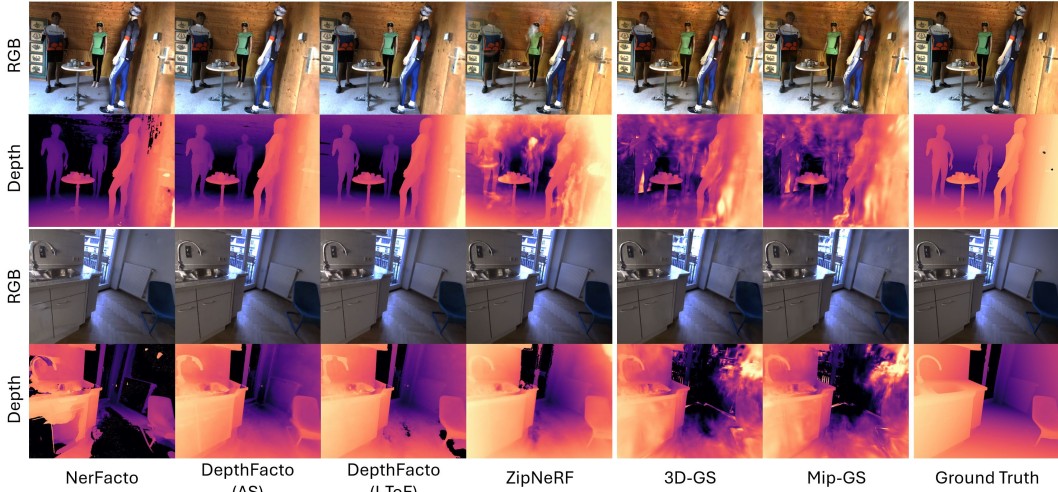

Figure 8: **Qualitative Evaluation of NVS Baselines.** We show rendering results of NeRF (NerFacto/DepthFacto, ZipNeRF) and Gaussian Splatting (3D-GS, Mip-GS) methods for both RGB and depth. The results show that realistic photometric appearance (RGB) does not always coincide with high quality depth rendering, while using sensor depth significantly improves the geometric understanding regardless of its modality. This indicates a larger room to improve geometric scene understanding with recent methods to allow for realistic real world vision application, such as Virtual Reality or Augmented Reality. Zoom in for details.

360 deg trajectories) compared to external tracking based pipelines like Hammer [9] or PhoCal [10] and more realistic scenes overcome the need of markers [28, 29]. We collect **two example scenes** to showcase the possible use of our pipeline. Examples are shown in Fig. 7 (c).

## 5 Benchmarks

In this section, we provide a benchmark for the two arguably most popular scene-level (e.g. train and test on the same scene) indoor 3D vision task, namely SLAM and Novel View Synthesis. We run all experiments on a RTX 4090 GPU and an i9 13th Gen CPU and show the result averaged over all scenes. Results on individual scenes can be found in the supplementary material.

**Novel View Synthesis Geometry Benchmarks.** We select four SoTA Novel View Synthesis (NVS) methods and depth supervision variants for a benchmark. We use NerFacto [30] with and without depth prior (Depth-Facto with two depth sensor priors), Zip-NeRF [31], Gaussian-Splatting [15] and Mip-Splatting [32]. Unlike any existing NVS benchmarks, our dataset provides highly detailed rendered ground truth depth without missing parts on the objects. This allows to further provide the evaluation metric on the depth rendering (i.e. RMSE, Abs Rel, Sqr Rel, $\delta < 1.25^n$) to evaluate the

Table 3: **Quantitative Evaluation of SLAM.** We benchmark three SoTA RGB-D SLAM methods, namely NICE-SLAM, CO-SLAM, and Gaussian-SLAM. We use an Active Stereo depth sensor (AS) and a ToF sensor (ToF) as well as the ground truth depth (GT) input to test the upper limit of each method.

| Evaluation | | Tracking | Mapping | | |
|---|---|---|---|---|---|
| Methods | Depth | ATE↓ [cm] | Acc↓ [cm] | Comp↓ [cm] | Comp Ratio↑ [%] |
| NICE-SLAM [34] | AS | 39.09 | 23.64 | 14.51 | 37.49 |
| | ToF | 24.02 | 21.60 | 14.63 | 46.18 |
| | GT | 5.47 | 2.91 | 5.91 | 77.11 |
| CO-SLAM [35] | AS | 39.19 | 32.38 | 12.95 | 40.07 |
| | ToF | 45.94 | 58.22 | 8.02 | 57.68 |
| | GT | 4.18 | 1.87 | 1.69 | 96.52 |
| Gaussian-SLAM [36] | AS | 38.20 | 37.94 | 11.74 | 41.36 |
| | ToF | 83.53 | 43.67 | 9.06 | 53.79 |
| | GT | 5.00 | 2.25 | 1.70 | 94.89 |

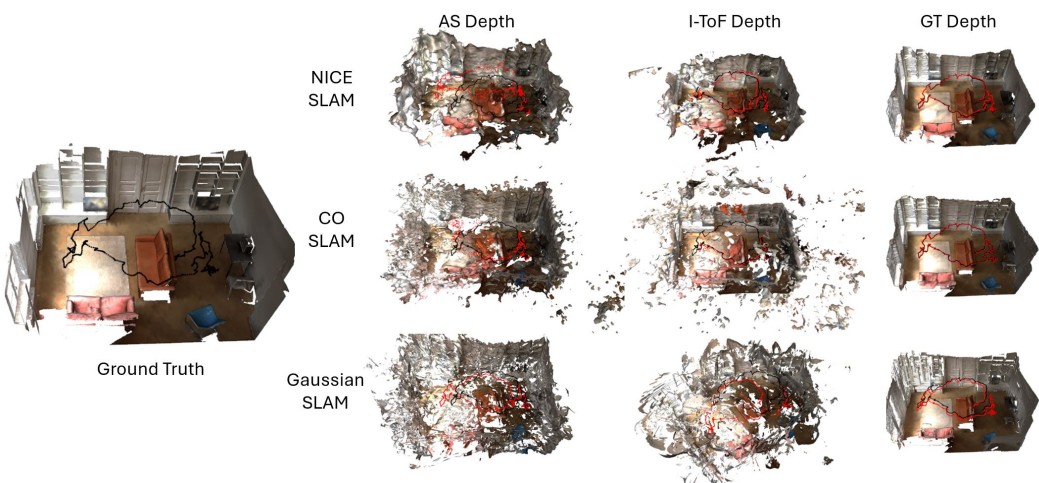

Figure 9: **Qualitative Evaluation of SLAM Methods.** Results of the 3 SoTA SLAM pipelines are shown as camera trajectory and scene reconstruction results. Each predicted camera trajectory (red) is compared with the ground truth camera trajectory (black). Zoom in for details.

geometric cue for each method together with traditional view synthesis metrics, such as PSNR, SSIM, LPIPS [33]. We use every 10th frame for training and the in between frame (every 10th frame + 5) for testing. Quantitative and qualitative evaluations are shown in Tab. 2 and Fig. 8.

**Indoor SLAM Benchmarks.** For SLAM, we select three SoTA RGB-D SLAM methods as benchmark, namely NICE-SLAM [34], CO-SLAM [35] and Gaussian-SLAM [36]. As aforementioned SLAM methods require depth as input, we run experiments on two sensor depths, Active Stereo and ToF, to evaluate the performance on a real-life scenario, as well as on rendered ground truth depth to understand the maximum possible performance of each method in the best case scenario. We use all frames to train and evaluate the obtained camera pose and scene reconstruction. Tracking performance is evaluated by absolute trajectory error (ATE) and RMSE [37]. Mapping performance is assessed using three common metrics for Neural RGB-D SLAM [38, 34, 35]: Accuracy, Completion, and Completion Ratio. Before mesh evaluation, unseen and occluded regions are culled using the strategy proposed in [39]. Quantitative and qualitative evaluations are shown in Tab. 3 and Fig. 9.

## 6   Limitation and Conclusion

SCRREAM constitutes an indoor 3D dataset annotation framework capable of annotating individual high quality scene meshes. We exemplify its use for annotation of four different tasks for which we provide example datasets. Furthermore, benchmarks on the reconstruction and SLAM dataset

using most recent methods are given. While the scene annotation is more detailed, the used hardware setup might be expensive, complex, and time-consuming, making the dataset annotation pipeline less scalable; for human reconstruction, 6d pose estimation dataset, we only include a few sample dataset scenes without benchmarks on the corresponding topics. To make individual images most comparable, we opted for a fixed exposure time which can cause under/over-exposed images and motion blur in darker scenes. Using a feature-based matching and pose estimation method requires static capturing conditions to ensure highly accurate camera pose estimation during the annotation stage. This limits the scene editing setup we propose is only capable of removing objects, not actual human interaction involved [40]. Limited by the scanning speed, we use mannequins instead of real humans that lack motions and realism, especially in the face and hand areas. Regardless of these drawbacks, our dataset is the only dataset to our knowledge with such an accurate setup covering the indoor room with a hand-held camera. This uniquely allows in-depth geometric evaluation and benchmarking of methods for most popular 3D applications such as NVS and SLAM. We strongly believe that our dataset and benchmark can bring forward the field through objective evaluation and public access for the research community by not simply taking sensor depth granted ground truth.

This work is from R&D Project at TUM funded by Huawei Research & Development (UK) Ltd.

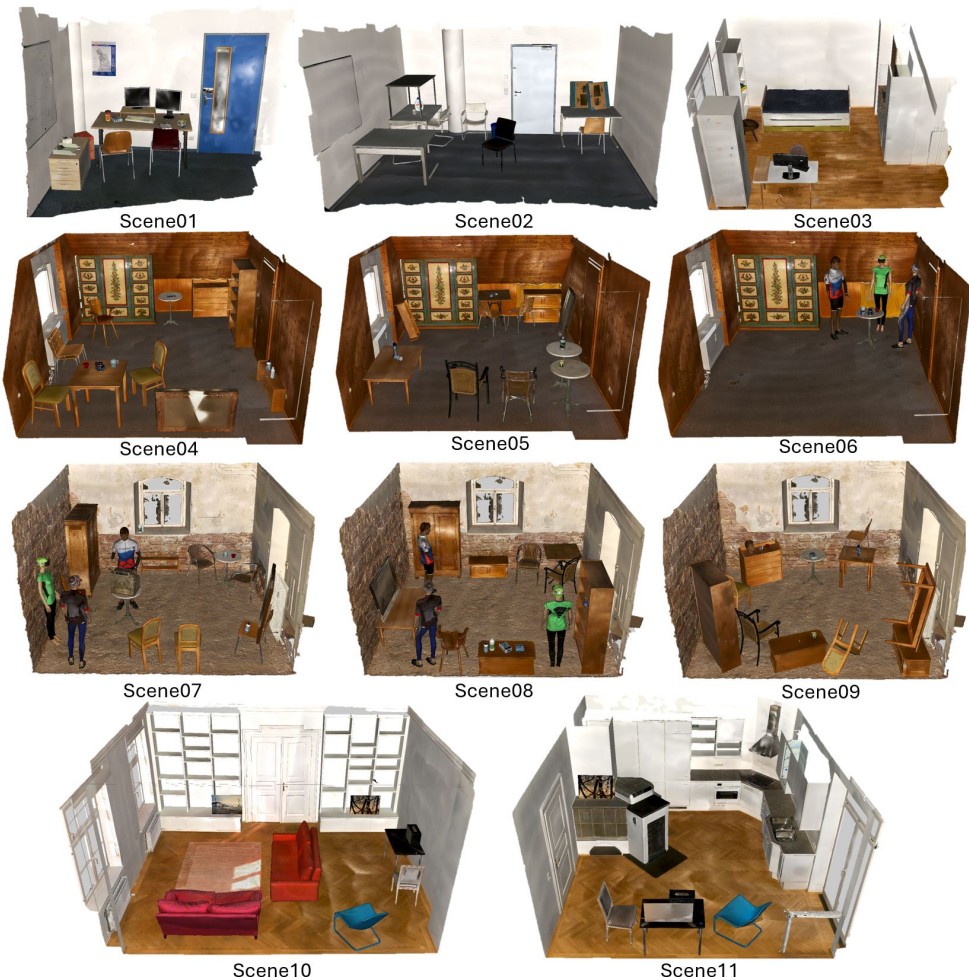

Figure 10: **Example of Indoor Reconstruction and SLAM Scenes.**

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
