# OpenReview forum: "SCRREAM : SCan, Register, REnder And Map: A Framework for Annotating Accurate and Dense 3D Indoor Scenes with a Benchmark"
_NeurIPS.cc/2024/Datasets_and_Benchmarks_Track — NeurIPS 2024 Track Datasets and Benchmarks Poster_

### Official Review · Reviewer_SLq9 · 2024-06-14
**Yet another dataset for NERF and GS with huge efforts**

**Rating:** 8
**Confidence:** 5
**Clarity:** It is well written.

**Review:**

Originality-wise, the dataset introduced by the author falls short. The concept is widely recognized and doesn’t address specific challenges in the field, such as the size of the area, introduction of degenerate environments, added noise, or other challenging conditions that are pertinent to practical applications.

Clarity-wise, the paper is quite easy to follow. It does well in explaining how the data is created, compared with other datasets, and how some popular algorithms perform when tested on this new dataset.

Quality-wise, it is evident that the author has invested considerable effort into the development of this dataset.

However, as someone more aligned with classical SLAM challenges and focused on addressing real-life problems like sudden light changes or degenerate environments, I find that this dataset does not meet those specific needs. It appears to be another option for those looking to implement GS or NeRF, rather than addressing the more acute challenges faced in dynamic real-world settings for wider SLAM community.

**Strengths:**

The author has demonstrated commendable diligence in data capturing and has rigorously tested the dataset against a set of baseline metrics.

**Additional Feedback:**

The dataset clearly reflects significant effort, and this is commendable. The writing style is clear and the documentation is thorough. However, as a submission for NeurIPS, the author needs to more clearly articulate the unique contributions of this dataset.

Usually, we set challenges for different algorithms to improve their performance. Currently, I do not see a significant challenge for the SLAM system in working with this dataset. It is small room and does not have issues with degeneracy or lighting challenge Like in UMA or OIVIO dataset. Real-world challenges, however, need to be framed within the context of how this dataset advances the field or addresses these problems in innovative ways, e.g., solar glaring into the room near sunset.

**Correctness:**

The method contribution claimed by the author is already well-known within the gaming industry. While the dataset has been constructed correctly, it is tested against only a small pool of baseline benchmarks

**Documentation:**

Documentation is the best that i have seen today so far. It is clear on how to use it.

**Ethics:**

No ethics issue detected

**Limitations:**

Despite the author's commendable effort in reality capturing, the results are somewhat suboptimal. Specifically, in Figure 4, the captured PC display exhibits a distorted light field, which does not occur in real-world image scenarios. Additionally, the size and complexity of the captured areas are limited.

**Opportunities For Improvement:**

Adding more area would help.
If scanning large building is too much, author might consider mixed reality where some object is virtually created and some is scanned.

**Relation To Prior Work:**

The paper is well-referenced, effectively citing previous work related to GS and NERF dataset

**Summary And Contributions:**

This paper introduces another replica-like dataset, opting to scan real-world objects into a game engine to render scenes. This method, commonly used by game designers to create digital twins, involves capturing real environments and integrating them into virtual worlds. While the authors claim this as a contribution, it is not a novel approach within the gaming industry.

Given Huawei's involvement, one might expect a wider range of environments, from small rooms to auditorium-sized spaces, to demonstrate scalability and robustness. However, the dataset does not yet reflect this diversity. Although the effort put forth by the author is commendable, the contribution lacks novelty in its current form.

---

> ### Author Rebuttal · Authors · 2024-08-17
>
> 1. SCRREAM vs Replica.
> The reviewer writes that the “paper introduces another replica-like dataset [...] [using] a game engine to render scenes”. This is not correct, we detail the difference here. Please also see our answer “1. SCRREAM vs Replica vs ReplicaCAD” to reviewer Nfyn and check this short video for a visual comparison: https://youtu.be/gHAVHgMcS-g
>
>
> The Replica dataset scans real scenes and renders RGBD images to provide training sets and evaluates against the reconstructed 3D digital twin. Thus both training and evaluation set is SYNTHETICALLY created (although from real data). Our dataset SCRREAM in contrast provides the REAL multi-modal RGBD trajectory as it would appear in a real use case including real sensor noise both on RGB and depth. Only the ground truth (e.g. depth map and semantics) are rendered from the highly accurate meshes using the registered camera pose.
>
> We create our ground truth data compositionally, i.e. we scan all objects, and compose the scene by registering them, thus providing fine details even on small scale objects and thin structures that have been pre-scanned. Replica, in contrast, follows a reconstruction paradigm (with post processing steps), thus it is not possible to capture small details. Rendering of 3D digital twin (as in Fig.4 (a)) is only an intermediate step to this register step where real camera images are co-registered with the scanned meshes / scenes. Note that this rendered RGB image allows the reader to see the high quality of reconstructed fine structures such as the legs of chairs that are missing in the Replica dataset. Fig.4 (b) describes the registration between the real image sequence and the synthetic image sequence to extract the real camera pose. Once the registration is done, we discard all the rendered RGB images (as we have REAL data) and keep only real RGBD images (Note that all RGB images are REAL in Fig.4 (b)).
>
> In this way, we can provide real RGBD images (Fig.4 (c) RGB / Active Stereo Depth, ToF depth) for which the sensor noise is also REAL, but geometric depth and semantic GT come from the pixel-perfect multimodal ground-truth registration that is way beyond what existing sensors can provide.
>
> As far as we know, this is the first dataset that features a free handheld moving camera in a REAL indoor scene, but overcomes the drawback of the existing dataset with incomplete meshes (cf. Reviewer v4H5’s comment) and provide synthetic dataset level quality of GT. The compositional nature of our dataset creation is the key factor to allow this. Note that for example ScanNet++ claims also high quality of meshes in their dataset, but we found that many meshes are missing parts and thin structures (cf. Fig. 14, supp. mat.). We opt for realism and quality over synthetic data and quantity. Missing structures such as the ones in ScanNet++ cannot be used to render depth for evaluation of the geometric understanding in Novel View Synthesis methods. Other construction-based datasets (such as ScanNet++ and Replica) do NOT provide the real sensor depth, and the depth has to be rendered from the mesh - thus comes at higher quality than commercial sensor, not providing real sensor noise and is perfectly view-consistent, all of which widen the sim-to-real domain gap for RGBD methods evaluated on this data. We believe that this attempt is indeed “pertinent to practical applications” and emphasize this point with the above argumentation more clearly in the paper now.
>
> 2. Dataset size and diversity.
> With SCRREAM we fill the void of a dataset that provides real data including real sensor noise together with highly accurate ground truth. For this, we design a new acquisition pipeline that opts for realism and quality over synthetic renderings and quantity. This comes at the cost of a more involved acquisition setup, but enables evaluation of pipelines closer to reality. We agree that it would be great to have an even wider variety of scenes and even more data. We detail the entire pipeline for anybody in the community to extend the dataset. Huawei’s support of this project made all of this uniquely possible given the risk involved to develop such an acquisition pipeline in the first place. Although the review process is single blind, we would kindly ask the reviewer to consider a fair and equal judgment of all papers irrespective of the author affiliations (re: “Given Huawei's involvement, one might expect a wider range of environments.”). Many thanks.
>
>
> 3. More challenges for the SLAM community.
> We agree with the comment that this data is even more suitable for novel view synthesis evaluation (NeRF and GS) than it is for SLAM regarding sudden light changes. Having sudden light changes could add great challenges to the SLAM field, but then the data would be disqualified for standard NeRF or GS training as the temporal consistency in RGB image is not guaranteed. However, our data includes challenging scenarios, where the camera is looking at the bright sun outside of a window, thus involving strong glare (check scene04_full_00 trajectory), or a room containing both underexposure and overexposure due to strong light in the low light condition (check scene11_full_00 trajectory) as well as sudden motion/jittering of the camera. Also having high quality geometry can eventually help evaluate SLAM-based reconstruction in the presence of REAL sensor noise on depth which is a more REALISTIC scenario and not possible with current datasets such as Replica that are perfectly view-consistent and do not provide real noise on depth. We are grateful for this comment and stress these challenges now more clearly in the paper.
>
> 4. Conclusion.
> We want to thank the reviewer for all comments that helped to stress even more the differences to previous datasets such as Replica and ScanNet++. We include the given argumentation in the paper and add details as supplementary video material to make this even more evident.

---

> ### Comment · Reviewer_SLq9 · 2024-08-17
> **/**
>
> It's a good explanation for some of my concerns. I`ll increase one point for this. Overall, it is a huge work; as someone who has published this kind of work before, I know how much effort it takes.
>
> However, the defect issue has still not been addressed. Like how the author proposes to address the unrealistic lighting issue of the display monitor. It seems to be misrendered. Also, this issue can be seen on the wall and on the door where color light field has clear distortion.

---

> > ### Author Rebuttal · Authors · 2024-08-26
> >
> > We are thankful for the swift comment.
> >
> > We agree that rendered images may contain unrealistic parts. However, the purpose of RGB rendering is just to provide the feature map to help the real RGBD image sequence register its camera pose to our mesh. After we obtain the real camera pose via feature matching, we discard all the rendered RGB images.
> >
> > So, the rendered images are not used as the actual dataset, and the results demonstrate that the feature map derived from the rendered RGB images is robust enough to accurately determine the camera pose in the real RGB sequence. Therefore, we believe this is not a significant issue.

---

> > > ### Comment · Reviewer_SLq9 · 2024-09-05
> > > **/**
> > >
> > > Sorry for the late reply, I was busy with another conference. It's good explanation. I`ll increase the point accordingly.  And again, thx for the great work

---

### Official Review · Reviewer_v4H5 · 2024-07-24
**/**

**Rating:** 7
**Confidence:** 3
**Correctness:** Yes
**Clarity:** Yes

**Review:**

Pros:
+ More Precise Framework.  SCRREAM offers a detailed pipeline that ensures high-quality, dense annotations by scanning individual objects and registering them to the scene. This approach mitigates the common issue of incomplete meshes in traditional datasets.The method ensures the generation of accurate ground truth data, including detailed depth maps, instance masks, and other 3D annotations, which are critical for evaluating dense geometry tasks.
+ Versatility: The framework is potentially adaptable to various indoor perception tasks, including indoor reconstruction, SLAM, scene editing. This versatility makes it valuable for multiple research directions. What attracts me most is the editing part, although the author didn't set up benchmark on this task. It would be good to see quantitative benchmark and reflection over this aspect.

Cons:
- Manual Registration Step: The initial registration step requires manual correspondence selection, which could introduce human error and affect the accuracy of the final annotations. Automating this process could improve the robustness and efficiency of the pipeline.

-Experimental Settings: The evaluation focuses on a limited number of scenes (eleven), which may not fully represent the diversity and complexity of real-world indoor environments. (1) Expanding the number of scenes could provide a more robust evaluation. (2) NVS part only focus on simple view interpolation and dense view training setting. It would be better to conduct more detailed study  over settings like sparse view training, and view extrapolation.（3）It is a bit strained to claim human reconstruction as -- (a) it is not applicable to real human actors in dynamic motion (b) this task not necessary associate with captured scene.

-Others: (1)Missing comparison with HouseCat6D in Table1, also there is no explanation for each dimension listed in the Table.(2) In Figure 2, the sub figures (a)(b) are exactly the same ones with Hammer work, (3)line92 -- how to select the pairs of corresponding points? What is the concrete criterion for this?

**Strengths:**

Please refer to the “Pros” in the Review section.

**Additional Feedback:**

Please refer to the Review section./

**Documentation:**

Yes

**Limitations:**

Yes

**Opportunities For Improvement:**

Please refer to the “Cons” in the Review section.

**Relation To Prior Work:**

Yes

**Summary And Contributions:**

This paper introduces SCRREAM, a framework for annotating dense 3D indoor scenes. The framework involves scanning individual objects, registering them to the scene, rendering realistic synthetic images, and mapping real image sequences to obtain accurate camera poses.  The authors provide benchmarks for SLAM and NVS Tasks using eleven example scenes, demonstrating the potential utility of their approach.

---

> ### Author Rebuttal · Authors · 2024-08-17
>
> We thank the reviewer for the positive feedback and agree that the “high-quality, dense annotations” can be “adaptable to various indoor perception tasks”. We want to thank for for identifying clearly that our “approach mitigates the common issue of incomplete meshes in traditional datasets” such as Replica.
>
> 1. Manual Correspondence Selection:
> The manual initial selection of correspondences is followed by an automatic ICP based refinement step. We empirically find the ICP based refinement to be very robust in our case as both our meshes contain a large quantity of point clouds / mesh faces / textures / and highly accurate surface normals. The manual correspondence search therefore does not negatively influence the automatic refinement except one willingly selects completely wrong correspondence on purpose. In practice to mitigate degenerate cases, we make sure to select points far away from each other and at least five points that are not laying on the same line.
>
> 2. Experimental Evaluation
> (1) Scene Quantity.
> With our dataset, we provide a set of diverse household scenes such as office, kitchen, living room, bedroom and basement, comprising unique challenges. We opt for accuracy and realism over synthetic rendering and quantity and provide a detailed description of the data acquisition process for others to expand in any possible direction which we exemplify in SCRREAM.
> We further kindly refer to our answer “2. Dataset size and diversity” of reviewer SLq9.
>
> (2) Sparser Training of NVS: We can provide NVS training with much sparser views on the selected scenes leveraging most popular current sparse NVS pipelines such as pixelNeRF, FreeNeRF, SPARF or InstantSplat. We can add these results to the final version of the paper (e.g. train with every 5th frame, 10th frame, 20th frames, etc).
>
> (3) Other applications such as Human Reconstruction & Object Pose Estimation: We agree that we only provide a small set of human scenes for semi-dynamic human reconstruction. However, our claim is that our dataset capturing framework can be extended to other fields beyond the Novel View Synthesis and SLAM, such as 6D Pose and Human reconstruction. We exemplify these use cases with some accurate labels in those domains and add the HouseCat6D comparison to table 1 as suggested. We want to thank the reviewer for this comment and stress this even more in the text to make it crystal clear.

---

### Official Review · Reviewer_Nfyn · 2024-07-25

**Rating:** 6
**Confidence:** 4
**Correctness:** Yes
**Clarity:** Yes

**Review:**

See strength and weakness. I think the dataset collect framework is novel and useful, but given the scale of the dataset and unclear advantage to previous dataset work, I would give borderline reject as my initial rating.

**Strengths:**

1. The work proposes an interesting approach for multi-step 3D reconstruction. The pipeline seems clever by scanning objects first and later do matching with the partial scan. The approach can cut down noise in depth and pose from real sensors by matching with the synthetic features.

Though the pipeline seems not that scalable, it is a good approach to construct smaller number and higher quality digital twins.

2. The work adopted several sensors and made multi-level scanning. The environment or wall scanning and partial object scanning both seems good-quality, which is a key for GT depth creation.

3. The paper has benchmarked several novel view synthesis works on the dataset, which is useful in understanding the performance.

**Additional Feedback:**

1. It is a bit weird when referring to structured light stereo camera, you use "D435" to mention it, but for ToF you use "ToF" but not its sensor make.

2. "Free-Hand Camera" is a weird term. Do you mean it is "hand-held and move freely" (free-moving hand-held, or 6DoF hand-held) or do you mean it does not need to move by hands (hand-free).

3. How dense is the Artec Leo scanning. Does the sensor do some post-processing so you can get watertight mesh, or does the sensor natively support highly dense mesh scanning?

**Documentation:**

Yes

**Ethics:**

No ethical consideration found

**Limitations:**

No negative societal impact

**Opportunities For Improvement:**

1. One of my main concern is the low number of scenes and how it compares with the existing dataset. There are only 11 scenes and the base environments/ objects are reused, though each scene is high-quality. I think one closely related dataset work is popular Replica and ReplicaCAD. Their purpose is to build a digital twin. Table 1 description is highly inaccurate. Replica is coupled with their render and simulator (Habitat) to render RGB sequence, camera pose, digital twin, instance segmentation, also editable, and free moving with 6DoF camera pose, and they later re-drew and CAD-lized the scene and rendered by Habitat simulator so I think the groundtruth is also accurate. (Their paper also mentions how they capture the data so the sensors shouldn't be -). (If the groundtruth depth/ pose is not accurate, a bunch of NeRF papers working on their depth and pose are not able to converge.)

They also provided a simulator for direct interaction with the CAD model objects. Overall, I may not be able to find that much advantage of this work compared with Replica/ ReplicaCAD, and the paper did not mention much about the comparison to Replica.

However, one thing I think useful for this work is more diverse objects and also include human reconstruction, but the number of scene is still few and scene contents are somehow repeated as my main concern to benchmark indoor novel view synthesis.


2. There is not much about sensor calibration details for the real sensor:

(1). How do you ensure the camera/ sensors are synchronized? Is there any mismatch in shutter trigger (software trigger)? (I assume the issue exists since they are separate sensors and do not cross-talk in the hardware-level synchronization

(2) Running COLMAP to get real camera pose is not very sound. COLMAP (or sfm) can fail or produce incorrect pose at textureless areas such as white walls or floor as shown in some cases in the figure. The best way to get pose is by IMU (which is how Replica collects pose and performs 3D reconstruction).

(3) I think it is a good idea to use multi-depth sensor to get finer depth for back-project to 3D, but how to consolidate depth values from different sensors (since different sensors have their advantage and weakness)? For example, a stereo camera says the point is 4.08m and ToF says it is 3.96m, how do you determine depth for back-projection for real image (i.e., in L114 what depth you are using here to get de-projected real image)?

(4) Did you calibrate sensors so that depth sensors can be correctly projected to RGB image and further used for back-projection? How?

3. About groundtruth pose and depth correctness: PSNR and SSIM in Table 2 are not that high (in comparison vanilla NeRF trained on Replica can attain 30-40 for PSNR), which might indicate the GT depth or pose might not be that correct so it prevents converging to better quality. Though it can be said the scene is more complex and challenging, I would like to see more proof or validation on the GT correctness.

**Relation To Prior Work:**

No. See Weakness

**Summary And Contributions:**

The work proposes a dataset with a novel strategy for building digital twins and collect 11 scenes. It also benchmark several novel view synthesis work and aims.

---

> ### Author Rebuttal · Authors · 2024-08-17
>
> We thank the reviewer for the encouraging comments, and we appreciate that the “dataset collect framework is novel and useful.” We agree that “higher quality digital twins” can be achieved through our “clever” scanning pipeline.
>
> SCRREAM Overview:
> SCRREAM provides multi-modal REAL sensor data, including sensor noise, along with highly accurate ground truth rendered from the SYNTHETIC digital twin. This approach prioritizes both accuracy and realism over quantity. Please see this video for a visual comparison:
> https://youtu.be/gHAVHgMcS-g
>
> 1. SCRREAM vs Replica vs ReplicaCAD:
> The Replica dataset is excellent for SYNTHETICALLY generating large quantities of RGBD data from scanned rooms. It is semi-real by design, with meshes obtained via 3D reconstruction methods. However, it has limitations with photometrically challenging materials and small structures, leading to over-smoothed meshes and missing details (e.g., Fig.15, supp. mat.). This results in SYNTHETICALLY rendered RGB trajectories being unrealistic, as they lack geometric detail and exhibit perfect multi-view consistency, leading to higher PSNR/SSIM metrics.
>
> Replica’s SYNTHETIC renders miss real sensor noise and produce perfect view-consistent depth, unlike real-world sensors that suffer from noise effects such as multipath interference and ambient light interference. Depth-based methods benefit from clean, consistent depth data, which is not reflective of real-life scenarios. SCRREAM’s REAL depth sensor data includes these realistic noise effects, removing sim-to-real gaps during evaluation, even though training is more challenging.
>
> ReplicaCAD is a FULLY SYNTHETIC dataset with 3D assets closely matching the original scenes, addressing mesh completion but sacrificing realism. This makes the sim-to-real domain gap more significant compared to Replica.
>
> In contrast, SCRREAM sequences are captured with real sensors, providing their complete data from each perspective. While challenging due to sensor noise, we offer fully registered camera poses and high-resolution meshes as reliable ground truth. This allows us to evaluate the methods with geometric understanding (depth) or semantic understanding and camera pose estimations accurately even with variations in RGB image quality due to real-world conditions, and furthermore allows us to evaluate real depth sensors inaccuracies using the GT depth renderings.
>
> 2. Sensor Calibration:
>
> (1) Synchronization: Achieved through hardware with wired signal triggers (lines 8-9, suppl.mat). All sensors support hardware synchronization, using a Raspberry Pi as the signal generator, as validated in our previous work (HouseCat6D, CVPR 2024, Sec.3.3).
>
> (2) Camera Poses / COLMAP: Although COLMAP may struggle in textureless regions, we verified camera pose quality using augmented rendered masks and depth, discarding sequences with non-converging trajectories. A video illustrating this can be found in the “video.avi” file per scene (cf. Fig.16, suppl. material).
>
> (3) Depth Warping: We align depth using the sensor’s own data, reflecting how real-world sensors operate. While alignment with rendered GT depth is possible, it would not reflect what users receive from actual sensors. Our approach aims to provide the most realistic noisy depth, in line with our focus on realism and accuracy over synthetic data.
>
> (4) Extrinsic Calibration: Each depth sensor is calibrated to the RGB camera via a long sequence focused on a static checkerboard, using synchronized images. We align trajectories between RGB and depth sensors using Horn’s method, saving the fixed 4x4 matrix offset as the extrinsic parameter. We run this calibration process between RGB<->ActiveStereo (D435), RGB<->ToF (Lucid Helios). For the depth sensors, Infra Red images are used to detect the checkerboard.
>
> 3. SSIM / PSNR Values:
> Lower SSIM and PSNR values in SCRREAM compared to Replica are due to more challenging scenes, with REAL lighting conditions and sensor noise. The semi-synthetic nature of Replica, combined with less detailed RGB renderings, eases image synthesis and boosts evaluation metrics. Our results in Table 3 (GT rows) show that training with SYNTHETIC GT depth significantly improves evaluation, which reflects the standard practice in Replica, where REAL sensor noise is absent. If our GT were not good, using GT depth would severely hamper the trajectory estimation as well as create incomplete reconstructions due to inconsistency. However, the training with GT depth is consistently improving the results across all three tested methods (NICE-SLAM, CO-SLAM, Gaussian-SLAM).
>
> 4. Conclusion:
> We appreciate the additional feedback, which has helped us improve the paper. We have emphasized the differences between SCRREAM and Replica and expanded on calibration and synchronization details in the supplementary material.

---

> > ### Comment · Reviewer_Nfyn · 2024-08-30
> > **Thanks for the comments**
> >
> > I appreciate the authors giving these details. I think the above discussion with Replica and ReplicaCAD should be put into the paper or supplementary with some visual comparison to best show the differences to explain why the proposed dataset is more advantageous over Replica and ReplicaCAD, especially since they are the most related and representative datasets. Without a visual comparison and explanation, readers may not easily agree with the mentioned points. During the rebuttal phase, the manuscript is editable, but the current manuscript seems to circumvent the comparison without mentioning Replica and ReplicaCAD much. The details of sensor calibration are also important technical details and should be included in the supplementary.
> >
> > Overall, I maintain my original score for the current form of the manuscript.

---

> > > ### Author Response · Authors · 2024-09-05
> > >
> > > We want to thank the reviewer for the comment. We follow the suggestion and add the additional discussion on both Replica(CAD) and sensor calibration to the supplementary material for a camera-ready version of the paper. PDF upload seems deactivated on our side.

---

> > > > ### Comment · Reviewer_Nfyn · 2024-09-06
> > > >
> > > > Thank you. I have raised my score. Please make sure the materials are properly included in the final version.

---

### Author Rebuttal · Authors · 2024-08-17

We want to thank all reviewers for their constructive and concrete feedback!
We appreciate reviewers find our “high-quality” (Nfyn, v4H5) “more precise framework” (v4H5) with “dense annotations” (v4H5) “novel and useful” (Nfyn). We agree that SCRREAM is “adaptable to various indoor perception tasks” (v4H5) and the “approach mitigates the common issue of incomplete meshes in traditional datasets” (v4H5) such as Replica and others. We appreciate that the "paper is quite easy to follow" and well explained (SLq9) and that "documentation is the best” (SLq9).

We want to emphasize that SCRREAM uniquely combines REAL multi-modal sensor data for RGB and depth with accurate ground truth, thus allowing us to utilize REAL data for training and optimization of methods. In contrast, many other approaches such as Replica use (semi-)SYNTHETIC rendered RGBD data without sensor noise for training . We opt for accuracy and realism with REAL data over quantity and simulation from SYNTHETIC renderings to imitate as closely as possible a REAL use case.

And we want to re-emphasie that our entire dataset is freely available with the download link provided on github. Feel free to download and check further details of the dataset (e.g qualitative videos or entire camera sequence) if there are further details to be checked during the discussion phase. Please find also a small visual comparison of this difference in this video ( https://youtu.be/gHAVHgMcS-g ) and details to the reviewer comments below.

---

### Decision · Program_Chairs · 2024-09-26

**Decision:**

Accept (Poster)

**Comment:**

The paper proposes a framework where individual objects are scanned, and then registered to compose an entire scene (the empty room, and room with objects are also scanned). The resulting dataset consist of 11 scenes (rooms) scanned in the above manner.  Experiments are conducted on the dataset to evaluate different methods for novel scene synthesis and indoor SLAM.  The paper indicates that the dataset can be used for object removal / scene editing, human reconstruction (with a posed mannequin), 6D pose estimation, but no experiments are conducted on these tasks.

Reviewers are appreciative of the immense effort that went into the data collection.  Although only 11 rooms are captured, the process is laborious and time-consuming, and results in scans of a furnished coupled with a decomposable scene (of room architecture + scanned objects).  The AC also finds the data to be unique and commend the authors for their effort.

However, the AC has the following concerns
1. Availability of the pipeline and insufficient details is provided to replicate the pipeline
   - The way the paper is presented currently highlights the scanning and dataset annotation framework as the main contribution of this work.
   - However, it is not clear to the AC whether an actual "framework" is provided and will be made available to the public. It is also not clear whether there is sufficient details provided to replicate the framework as there is little information provided for the registration step and reviewers also had questions (Nfyn)
2. Small size of proposed dataset (Nfyn, v4H5)
3. Experiments does not utilize key features of the dataset and can be applied to other datasets as well

Given the positive ratings from reviewers, the AC believe that the work could be of interest to the community despite its shortcomings.

The authors are strongly urged to improve their paper based on feedback from the reviewers and the AC:
1. Provide missing details about the framework and data collection process (Nfyn)
2. Clarify whether an actual full-framework will be provided or revise the paper to indicate that a data creation "process" is described. If a full-framework is claimed, the authors are strongly encouraged to provide appropriate code.
3. Add discussion about the scalability and limitations of the proposed approach for scanning environments
4. Add discussion of recent work ParaHome [1] that scans individual objects and then captures humans interacting with the objects in scenes
5. Add discussion about differences to prior datasets with synthetic/real pairings such as Replica/ReplicaCAD, RoboTHOR[2] scenes, etc.
6. Make clear that scene editing, human reconstruction, 6D pose estimation are potential use cases for the dataset that was not actually tested.  Please also tone down claims of appropriateness of the dataset for investigating human reconstruction

[1] ParaHome: Parameterizing Everyday Home Activities Towards 3D Generative Modeling of Human-Object Interactions [Kim et al. 2024], https://arxiv.org/abs/2401.10232
[2] RoboTHOR: An Open Simulation-to-Real Embodied AI Platform [Deitke et al. 2020], https://arxiv.org/abs/2004.06799